# Fractional Order PI-Based Control Applied to the Traction System of an Electric Vehicle (EV)

**German Ardul Munoz-Hernandez** *,†, **Gerardo Mino-Aguilar** †,
**Jose Fermi Guerrero-Castellanos** † **and Edgar Peralta-Sanchez** †

Benémerita Universidad Autónoma de Puebla (BUAP), Facultad de Ciencias de la Electrónica,
Ciudad Universitaria, Puebla 72570, Mexico; gerardo.mino@correo.buap.mx (G.M.-A.);
fermi.guerrero@correo.buap.mx (J.F.G.-C.); edgarps@ieee.org (E.P.-S.)

* Correspondence: gmunoz@ieee.org; Tel.: +52-222-255-8932
† These authors contributed equally to this work.

**Abstract:** This paper presents a design of a cruise control based on a fractional-order proportional and integral (PI) direct torque control applied to the traction system of an electric vehicle (EV). The paper also discusses the modeling, control design and simulation, resulting in a numerical simulator composed of detailed models of the main components: transmission system, induction motor, power electronics, control system, and vehicle dynamics. The simulation was developed in MATLAB/Simulink and will allow the estimation of the energy consumption of an EV under specific configurations. Simulation results show the efficiency of the designed control. These simulations were carried out using the velocity profiles given by the New Europe Drive Cycle (NEDC).

**Keywords:** electric vehicle (EV); traction system; DTC; dynamic simulation; fractional order control

---

## 1. Introduction

Among the many problems that have been considered important worldwide the two that stand out are pollution and overpopulation. Overpopulation has created high demands for transport, both short and long-distance, which increases the problem of pollution [1].

Electric motors consume half of the electric power generation, so increasing their efficiency impacts the saving of electric energy. Different control techniques that use digital systems (as well as new power electronics devices) have evolved in the works looking for improving the performance of electric motors [2].

These electric machines have also made progress in their design. Three-phase induction motors (IM) are normally used in the industrial sector, due to their simple design, high reliability and low cost. Frequency converters are traditionally used to control the speed and stop of induction motors with precision [3]. The control of the speed of the motors allows the saving of electrical energy. Therefore, the use of variable speed drives has been increased to control not only the speed, but also the torque, flow and position.

The majority of the vehicles that circulate around the world nowadays consume fossil fuels. These vehicles use internal combustion engines (ICE), with a performance of the transformation of energy from the fuel tank into mechanical energy of less than 30%, which represents a great loss. They also produce $CO_2$ and $NO_x$ emissions into the atmosphere, which leads to a large localized pollution in large cities. In recent years, electric vehicles (EV) and hybrid electric vehicles (HEV) have proven to be an alternative as they are a clean, effective and ecologically friendly transportation system [4].

In an electric vehicle, its movement is provided by an electric motor that has batteries as its primary source of energy. Excluding the generation of electricity, EVs do not generate $CO_2$ and $NO_x$

emissions, allowing buffering and presenting a part of the solution to the problem of global warming. In addition, they stand out for their high performance in the transformation of the electrical energy of the battery into the mechanical energy with which the vehicle will move (60–85%) [5]. However, EVs present two main problems that must be addressed and resolved, they have low autonomy and long recharge times. These problems are closely related to the vehicle's battery bank. The batteries must have enough energy for the vehicle to make a certain journey and can accelerate and decelerate if necessary [6].

Mathematical models are important in estimating the energy consumption of electric vehicles [7,8], since they allow us to numerically simulate the behavior under different conditions and scenarios. The design or implementation of each component (e.g., transmission system, electric motor, power electronics and battery) is a delicate task since the power levels of one component can affect that of others, making the vehicle unnecessarily expensive or inefficient.

The present work covers the process of modeling, control and simulation of an EV. The motor used for the traction system is an induction motor (IM) of 1.1 kW. Then, the dimensions and characteristics of each element of the EV (batteries, voltage converters, traction and control systems) were selected according to the characteristics of the IM, together with the following minimum specifications: maximum speed without slope of 30 km/h (maximum speed allowed within the campus university, CU-BUAP) and with a total mass (including the driver) of 200 kg. Special attention was paid on the development of IM speed control, which allows implementing a speed cruise control for the EV. This control is done through the well-known direct torque control (DTC) and a proportional and integral (PI) control of fractional order.

DTC is an approach employed to regulate electric motors. This technique selects an appropriate voltage vector to regulate torque and flow in electric motors. That vector is chosen from a switching table, which is predefined. The above is accomplished by computing instantaneous torque and flow, using stator status information. By selecting the optimal switching states of the inverter, torque and flow are regulated independently and directly. Hysteresis controllers are used to limit the flux and torque errors. Because only stator resistance is used to implement the DTC, the implementation is simple. For this work, a DTC was designed in MATLAB/Simulink to evaluate the performance of an induction motor operating at different load and speed conditions, the IM model and the inverter were also designed in the referred software.

Interest in the application of fractional order algorithms in cruise control has been manifested recently, for instance, Hosseinnia et al. [9] demonstrated the effectiveness of a fractional hybrid control at low speeds, by applying this in a real circuit. Kumar et al. [10] presented a paper where a hybrid electric vehicle is controlled using a fractional order fuzzy proportional, integral and derivative (PID) scheme, they reached a very robust behavior for speed control of the vehicle. Taymans et al. [11] have applied a fractional linear feed-forward plus a pre-filtering to cruise control of an electric vehicle.

The content of this paper is organized as follows. First, a brief description of the dynamics of the vehicle, seen as a rigid body, is presented. Then, the traction system where each of its components is briefly explained, followed by the description of the design of a cruise control and analyses of the dynamic behavior and energy consumption through the simulation of a management cycle. Finally, some conclusions and perspectives are presented.

## 2. Electric Car Model

### 2.1. Mechanical Model

In order to develop a mathematical model that represents the dynamics of the vehicle, a balance of forces is made. Let $v_v(t)$ be the velocity of the EV, $m_v$ its total mass, $F_t(t)$ the tensile force generated by the contact of the tires with the road and $F_p(t)$ the force of disturbance due to the aerodynamic drag, the friction of bearing and gravity. These forces are shown in Figure 1.

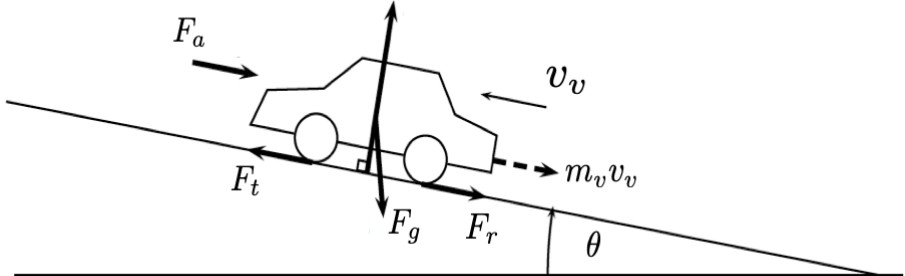

**Figure 1.** Diagram of forces acting upon the vehicle.

Applying Newton's second law, we obtain the differential equation that relates the tensile force $F_t(t)$ with the speed $v_v(t)$ of the vehicle:

$$m_v \dot{v}_v(t) = F_t(t) - F_p(t), \tag{1}$$

where

$$
\begin{aligned}
F_p(t) &= F_a(t) + F_r(t) + F_g(t) \\
F_a(t) &= \frac{1}{2} \rho_a A_f C_d \left( v_v(t) + v_{wind}(t) \right)^2 \\
F_r(t) &= \mathrm{sgn}\left( v_v(t) \right) m_v g \cos\left( \theta(t) \right) C_r \\
F_g(t) &= m_v g \sin\left( \theta(t) \right).
\end{aligned} \tag{2}
$$

$C_d$ is the drag coefficient, $C_r$ is the coefficient of friction of the bearing, $A_f$ represents the frontal area of the vehicle, $\rho_a$ the density of the air at 20 °C, $\theta$ the angle of the slope, and $g$ the gravity acceleration.

The force $F_t$ is generated by the electric motor, whose generated torque is a function of the current flowing through its windings and its rotation speed. From (1), it is clear that the electric motor of the vehicle, together with the transmission system must be able to overcome the forces due to gravity, wind, friction and inertial effects.

### 2.2. Traction System

Figure 2 shows each of the components and subsystems that make up the EV and of which the mathematical model was obtained with the objective of implementing the computational simulator. In the following sections a description of the elements is given and the mathematical model of each of them is presented.

### 2.3. Transmission

Equations (3) and (4) can be used to calculated angular velocity, torque and power of the transmission system (Figure 2).

$$
\begin{aligned}
\tau_t(t) &= F_t(t) r_\omega, \quad \tau_\omega(t) = \frac{\tau_t(t)}{2} \\
\omega_\omega(t) &= \frac{v_v(t)}{r_\omega}, \quad p_t(t) = F_t(t) v_v(t),
\end{aligned} \tag{3}
$$

where $\tau_t$ and $\tau_\omega$ are the total traction torque and torque on each wheel, respectively, whose radius is $r_\omega$. $\omega_\omega$ is the angular velocity and $p_t$ the tensile power. In addition, it is assumed that the transmission

efficiency is given by $\eta_{TS} = 0.95$ and there is a velocity reduction factor $\gamma$. Consequently, the torque, angular velocity and power of the electric motor are given by:

$$\tau_m(t) = \frac{\tau_t}{\eta_{TS}\gamma} \quad \text{if } p_t \geq 0 \quad \text{and } \tau_m(t) = \eta_{TS}\frac{\tau_t}{\gamma} \text{ if } \quad p_t < 0$$

$$\omega_m = \gamma\omega_\omega$$

$$p_m(t) = \tau_m\omega_m. \tag{4}$$

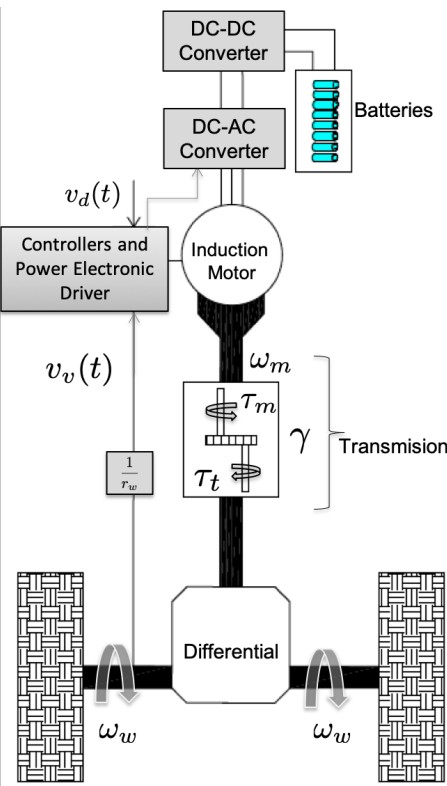

**Figure 2.** Subsystems that make up the electric vehicle.

**Remark 1.** *With the information given above, it is clear that the moment of total inertia seen in the frame of reference of the engine is given by $J_t = J_M + \frac{J_v}{\gamma^2}$, where $J_v = Mr_\omega^2 + J_\omega$ is the moment of inertia of the vehicle, with $J_\omega$ the moment of inertia of the wheels.*

**Remark 2.** *The nominal torque of the engine is 7.4 Nm, which is not enough to start the vehicle, so $\gamma = 6$ was chosen, enough for the EV to obtain a maximum speed of 30 km/h on flat terrain and a maximum speed of 15 km/h with 5° of slope.*

## 3. Direct Torque Control (DTC)

### 3.1. Direct Torque Control Strategy

Induction motors can be driven in many ways. From the simplest controller voltage/frequency approach to complex strategies like DTC or Vector control. The main differences between them are the motor's performance and the viability and cost of its real implementation. The first method, known as scalar control, applies a constant V/f relation, this approach does not use the rotor speed as feedback variable. Although it is a very simple method, its main disadvantage is the fact that the speed and torque response do not have good precision, because the stator flux and torque are not

directly controlled. In contrast, the vector control method introduces both the torque and flux control independently, using vector transformation. By using this method, the control accuracy increases significantly. However, its implementation could require enormous computational capability in the digital processor, and high accuracy in motor parameters estimation. DTC offers a fast response in torque and a higher dynamic performance, because DTC uses a simpler model than vector control.

The main objective of DTC is to calculate the values of flux and torque, in a particular instant of time, by using the stator variables of induction motor. The torque and flux are directly and independently controlled by selecting the optimal inverter commutation state and limiting the torque and flux errors via the torque and flux hysteresis controllers.

The main characteristics are [1–3]:

- Direct stator flux control and direct torque control
- Indirect regulation of stator currents and voltages
- Sinusoidal stator fluxes and stator currents approximation
- High dynamic performance even at locked rotor

The advantages of this approach are:

- Absence of co-ordinate transform
- Absence of voltage modulator block
- Minimal response time, even better than the vector controllers

However, possible problems could be presented such as starting requirement of flux and torque estimator. Also, a parameters identification is needed to reduce the inherent torque and flux ripples. A block diagram of the DTC is illustrated in Figure 3. As can be seen, the DTC is comprised by an estimator of the electromechanical torque, the stator flux, sector location, vector $I_s$ calculator, comparators, hysteresis blocks and a switching table, all of which elements are programmed in a Matlab/Simulink block.

The dynamic behavior of an induction motor is complex due to the coupling between the stator and rotor phases, where the coupling coefficients vary with the rotor position. Therefore, the machine is modeled by a set of differential equations with variable coefficients. A thorough knowledge of the induction motor model is vital to verify the effectiveness of the control algorithm proposed in this paper. The mathematical model of the rotor induction motor short-circuited in dynamic regime, in oriented field, coordinates are represented by a non-linear system of differential equations. To obtain the model, different aspects are considered. Below are the differential equations of the rotor flow belonging to the electric dynamic model of an MI based on the stator currents and the rotor time constant and the speed in the rotor. Since the rotor currents cannot be accessed directly, it is necessary to represent the model by means of the concatenated flow of the rotor from its projections on the d and q axes [12].

Calculation of the space vector of the stator current ($I_s$) can be done by using $i_a$, $i_b$, $i_c$ (motor currents) as shown in (5).

$$I_s = \frac{2}{3}(i_{an} + i_{bn}e^{j\frac{2}{3}\pi} + i_{cn}e^{j\frac{4}{3}\pi}), \tag{5}$$

where $I_s$ in d-q system coordinates is like (6)

$$I_s = i_{ds} + j i_{qs}. \tag{6}$$

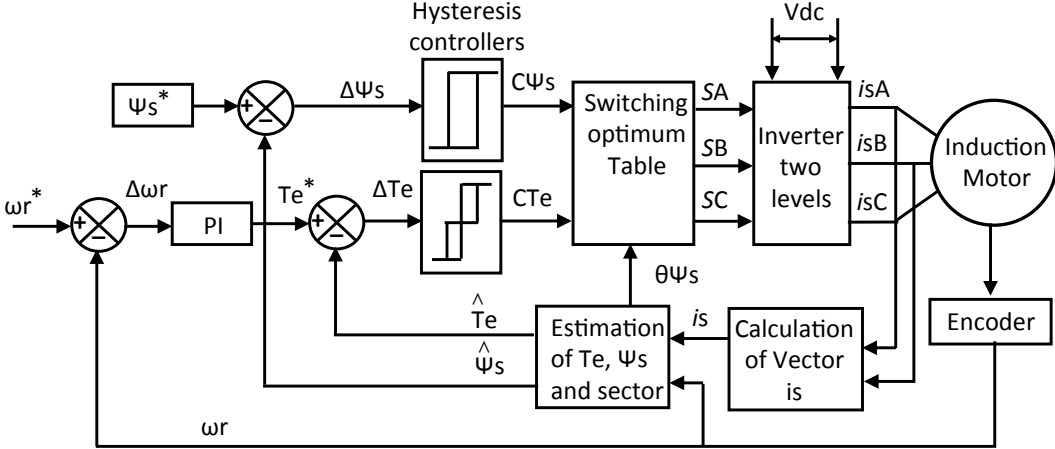

**Figure 3.** Current model direct torque control (DTC).

The stator flux ($\Psi_s$), the electromagnetic torque ($T_e$) and the sector location of the flux for the induction motor model are estimated in a block designed for that purpose. Equations (7) and (8) define the model used.

$$\frac{d\Psi_{dr}}{dt} = \frac{L_m}{L_r}i_{ds} + \omega_r\Psi_{qr} - \frac{1}{T_r}\Psi_{dr} \tag{7}$$

$$\frac{d\Psi_{dr}}{dt} = \frac{L_m}{L_r}i_{qs} - \omega_r\Psi_{dr} - \frac{1}{T_r}\Psi_{qr}. \tag{8}$$

The rotor flux and stator currents are used to estimate the electromagnetic torque ($T_e$) defined by (9).

$$T_e = \frac{3}{2}\frac{P}{2}\frac{L_m}{L_r}(\Psi_{dr}i_{qs} - \Psi_{qr}i_{ds}), \tag{9}$$

where $T_r$ (rotor constant) is given by:

$$T_r = \frac{L_r}{R_r}. \tag{10}$$

The stator flux ($\Psi_s$) in the stationary frame can be obtained by (11) and (12).

$$\Psi_{ds} = \Psi_{dr}(\sigma L_s) \tag{11}$$

$$\Psi_{qs} = \Psi_{qr}(\sigma L_s), \tag{12}$$

where $L_s$ is the stator inductance and $\sigma$ is the total link factor, which is determined by (13).

$$\sigma = 1 - \frac{L_m^2}{L_s L_r}. \tag{13}$$

From the $d$ and $q$ flux components the flux angle ($\Delta\Psi_s$) can be calculated as follows:

$$\Delta\Psi_s = arc(\frac{\Psi_{qs}}{\Psi_{ds}}). \tag{14}$$

A basic situation to be considered in the DTC is that the position of the stator flux vector must be known and must be precise, in order to cause the desired effect on the torque and flux [2]. This could be achieved by dividing the d–q plane into six sectors of 60 degrees each (4), in this way the sector in which the stator flux vector is located is known.

On the other hand, in situations where the speed is very low, vibrations can occur in the motor. This is mainly due to the presence of the increase in torque ripple, this can be resolved by reducing the

bandwidth in the hysteresis regulator of the stator flux, thus, the distortion is not increased during the change of sector, this is possible at constant torque [13]. More generally, the reduction in torque and flux ripple can be resolved when the DTC is performed by the analog method by reducing the hysteresis band of the torque and flux controllers. Since most modern systems use digital control due to its advantages. It should not be forgotten that digital algorithms need a runtime, which results in a delay when two consecutive signal samples are processed. Depending on the time delay, when the torque increases it can exceed the upper limit, defined by Hte+, resulting in an overflow or overreach. In the same way, when the torque is decreasing, it can reach values below the lower limit, defined by HTe-, producing a sub-range. Consequently, a great torque ripple is produced, deteriorating the behavior of the DTC. If the sampling frequency is lower, the ripple will increase [14].

Also, although the DTC offers good dynamic behavior and fast torque response, the starting current reaches high values. This problem can be solved by adding a closed loop of the magnitude of the stator current vector IS, in that loop the zero vector is applied if the current reaches the maximum limit determined by a current hysteresis controller. When the current exceeds its minimum limit, a vector determined by the switching table of the DTC is chosen [13]. There are other methods to solve the disadvantages in the DTC, which are applied by modifying the conventional method, such as:

1. The use of very fast digital control boards that allow the sampling frequency to be raised to high values (greater than 20 kHz).
2. Instead of applying only one vector during the sampling period, a duty ratio control method can be used in which two vectors are applied, one active during a percentage of the sampling period and the other, the zero vector, which is applied during the rest of the period to reduce the torque ripple. Since there is no linear relationship that governs this operation, fuzzy logic can be applied to solve it [15].
3. Predictive torque control can be applied with the help of spatial vector modulation (SVM). In this method, the appropriate vector and its duration are predicted to be applied during the next sample [16,17].
4. Inject a small but high frequency signal (>20 kHz) into the torque controller. A signal can also be injected into the flux controller to reduce the flux ripple. But the drawback of this method is that the switching frequency of the inverter must be increased, thereby increasing the switching losses. This method is suitable for a quasi-resonant type inverter [18].
5. Use a torque controller whose output is 1 or −1.

In this method active vectors are applied instead of zero vector to obtain rapid change in the magnitude of the torque. The advantage of this method is to avoid distortion in the flux path of the stator. Although the last two methods are effective in improving DTC behavior, in many cases high switching frequency values are obtained that are not acceptable. Therefore, the realization of the DTC with these methods is more suitable for drives powered by a quasi-resonant inverter due to its availability to work at high switching frequencies without increasing losses through power switches [14].

*3.2. Sector Location*

The torque reference regulating the magnitude and rotation of the stator flux is used to adjust the motor torque. Depending on the position of the stator flux space vector, the DTC defines six operating sectors. Once the angle and magnitude of the stator flux space vector are known, the DTC method needs to know the sector where the flux vector is located, which corresponds to one of the six sectors into which the stationary plane $d_q$ is divided. Any of these six control areas has a width of $\frac{\pi}{3}$, these are given by (15).

$$\frac{(2n-3)\pi}{6} \leq sector(n) \leq \frac{(2n-1)\pi}{6}, \tag{15}$$

where $n$ is the number of sector, ranging from 1 to 6.

Once $\Delta\Psi_s$ is known the sector can be calculated by using Figure 4. The sector provides information about the location of the voltage space vector of the inverter to place the voltage space vector of the inverter to control the flux and torque.

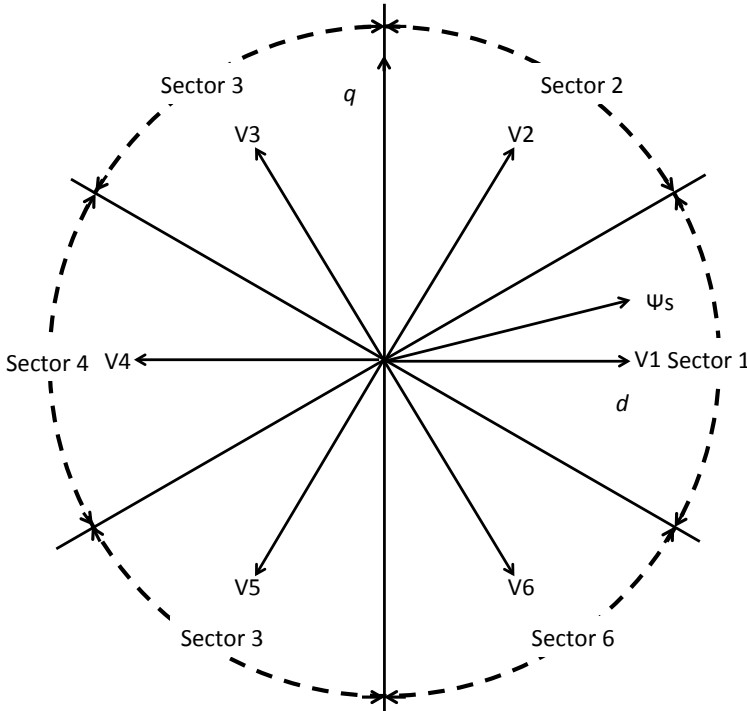

**Figure 4.** Voltage vectors and sectors definition.

The error $\Delta\Psi_s$ is processed by a two levels block of hysteresis, with a lower boundary defined by $H\Psi_s-$ and an upper boundary defined by $H\Psi_s+$. The band of the hysteresis controller is limited by two levels as shown in Figure 5, $H\Psi_s = (\Psi_s+) + (\Psi_s-)$. The stator flux is maintained within the hysteresis limits, these limits define the hysteresis band. The narrowness of this band determines how close the stator flux will be to the reference.

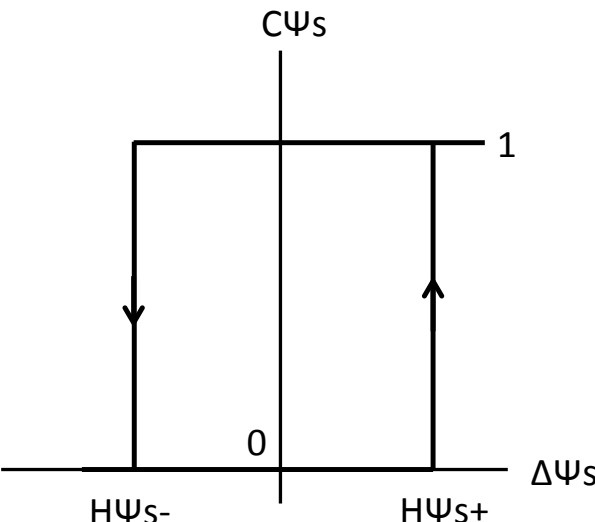

**Figure 5.** Two-level hysteresis controller.

The error $\Delta T_e$, resulting of subtracting the estimated from the reference torque, is managed by a hysteresis block of three-level, with the following error limits: $HT_e-$ and $HT_e+$. The loop of hysteresis $HT_e = (HT_e+) + (HT_e-)$ is shown in Figure 6. The purpose of the table for optimal switching is to deliver eight voltage vectors that regulate the amplitude and frequency of the inverter voltages supplying the motor (see Table 1).

By adjusting the amplitude and frequency, the torque and flux in the motor are controlled. Taking the state of flux and torque as a basis, Table 2 shows the voltage vectors for each sector.

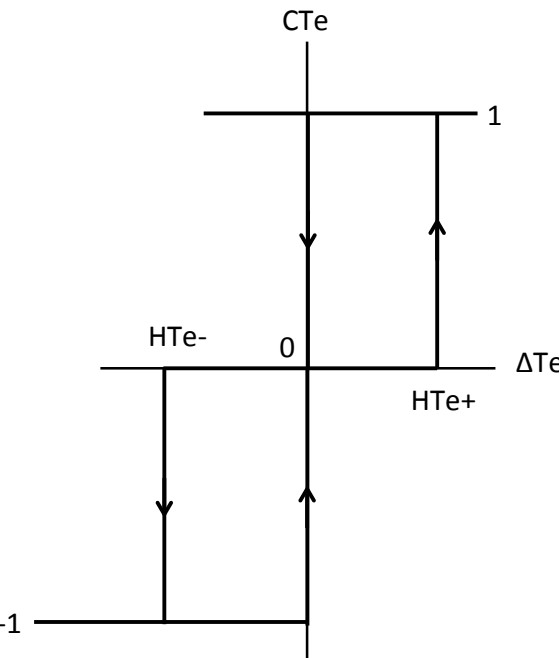

**Figure 6.** Three-level hysteresis controller.

**Table 1.** Possible states of the inverter.

| VSI State | $S_a$ | $S_b$ | $S_c$ |
|-----------|-------|-------|-------|
| V1 | 1 | 0 | 0 |
| V2 | 1 | 1 | 0 |
| V3 | 0 | 1 | 0 |
| V4 | 0 | 1 | 1 |
| V5 | 0 | 0 | 1 |
| V6 | 1 | 0 | 1 |
| V7 | 1 | 1 | 1 |
| V8 | 0 | 0 | 0 |

**Table 2.** DTC logic switching table.

| $C\Psi_s, CTE, \Delta\Psi_s$ | | $\Delta\Psi_s$ (1) | $\Delta\Psi_s$ (2) | $\Delta\Psi_s$ (3) | $\Delta\Psi_s$ (4) | $\Delta\Psi_s$ (5) | $\Delta\Psi_s$ (6) |
|---|---|---|---|---|---|---|---|
| | CTE = 1 | V2 | V3 | V4 | V5 | V6 | V1 |
| $C\Psi_s = 1$ | CTE = 0 | V7 | V8 | V7 | V8 | V7 | V8 |
| | CTE = −1 | V6 | V1 | V2 | V3 | V4 | V5 |
| | CTE = 1 | V3 | V4 | V5 | V6 | V1 | V2 |
| $C\Psi_s = 0$ | CTE = 0 | V8 | V7 | V8 | V7 | V8 | V7 |
| | CTE = −1 | V5 | V6 | V1 | V2 | V3 | V4 |

To control the switching state of the inverter, the DTC uses the eight voltage vectors defined in Table 1.

The rapid response of the DCT generates high currents that may activate the inverter protections. To cope with this inconvenience the stator currents are monitored to generate a closed-loop which handles the switching table sending a vector 0 whenever the current vector is over the allowed limit.

## 4. Fractional Order Control

Different variations of PID, including nonlinear approaches, have been reported in the literature. For instance, PID controllers with integral and derivative of fractional order have been published. Those works show an enhancement on the functioning of the controllers [19]. A generalization of the PID controller was proposed by Podlubny [20,21], that approach includes an integrator of order $\lambda$ and a differentiator of order $\delta$ ($PI^\lambda D^\delta$ controller). The transfer function of this controller is shown in (16).

$$C(s) = \frac{U(s)}{E(s)} = K_p + K_i s^{-\lambda} + K_d s^\delta \tag{16}$$

where

$\lambda > 0$, real integral operator.
$\delta > 0$, real differential operator.
$K_p$, proportional constant.
$K_i$, integration constant.
$K_d$, differentiation constant.

The fractional-order controller comprises three connected parts [22].
In discrete time the controller is:

$$C(z) = \frac{U(z)}{E(z)} = K_p + \frac{K_i}{\left(\omega(z^{-1})\right)^\lambda} + K_d \left(\omega(z^{-1})\right)^\delta. \tag{17}$$

In Equations (16) and (17), a PID controller in its classic form is conceived, if $\lambda = 1$ and $\delta = 1$. Otherwise, if $\lambda = 1$ and $K_d = 0$, then a PI controller is obtained. Actually, these controllers are derived from the fractional order control, which, due to its flexibility, allows to better adjust the dynamic activities of a system [19,23]. In this work a fractional-order PI was tuned to minimize the integral time-weighted absolute error (ITAE).

A usual method to implement fractional order parameters is by approximation. An approximation that has been largely studied is to implement the $S^q$ operators, where q is a non-integer, as a Lead-Lag compensators (18) [24].

$$\frac{V_o(t)}{V_i(t)} = \frac{1}{S^q} = \frac{(1-q)S + (1+q)}{(1+q)S + (1-q)} = A\frac{S + \frac{1}{A}}{S + A}, \tag{18}$$

where

$$A = \frac{1-q}{1+q}. \tag{19}$$

In the case of the derivative factor $A = \frac{1-q}{1+q}$.

In order to facilitate the implementation of fractional order derivate and integral operators, a discrete representation by Z transform could be used for the PID (20).

$$C(z)\frac{U(z)}{E(z)} = K_p + K_i\left(\frac{K_{foi}(z - Z_i)}{z - P_i}\right) + K_d\left(\frac{K_{fod}(z - Z_d)}{z - P_d}\right), \tag{20}$$

where

$P^i$ is the pole of the lag compensator

$Z^i$ is the zero of the lag compensator

$P^d$ is the pole of the lead compensator

$Z^d$ is the zero of the lead compensator

As all discrete implementation the sample time selected is crucial. If the Equation (20) is developed, the fractional order PID (FO-PID) can be seen as a second order filter, with two real poles and a pair of complex conjugate zeros (21). Therefore, techniques that allow fixing poles and zeros of compensators can be used to tune FO-PID's . For this work, the control parameters were selected using Root Locus and the method proposed by Awouda and Mamat [25]. The poles and zeros of the FO-PI controller are located depending on the dominant closed-loop pole desired.

$$\frac{U(z)}{E(z)} = \frac{z^2(K_p + K_{I1} + K_{D1}) + z(-K_p(P_i + P_d) - K_{I1}(Z_i + P_d) - K_{D1}(Z_d + P_i)) + K_p P_i P_d + K_{I1} Z_i P_d + K_{D1} Z_d P_i}{(z - P_i)(z - P_d)}, \tag{21}$$

where

$$K_{I1} = K_I K_{foi} \quad and \quad K_{D1} = K_D K_{fod}. \tag{22}$$

In this work the poles and zeros of the FO-PI are located looking for minimizing the ITAE index.

## 5. Statement of the Problem

Considering the equation that governs the dynamics of the EV, given by (1) in conjunction with the equations given for the transmission system (3) and (4). Then the dynamic of the vehicle is given by:

$$m_v \dot{v}_v(t) = \frac{\eta TS\gamma}{r_\omega} \tau_m(t) - F_p(t). \tag{23}$$

It was identified that in the Equation (23) the velocity of the car $v_v$ is the state, $\tau_m(t)$ is the control, and $F_p(t)$ is a disturbance. The objective is to develop a control strategy that allows maintaining the velocity of the EV at a desired constant speed, even in the presence of disturbances, for instance by finding a control signal $\tau_m(t)$ such that $v_v(t) \Rightarrow v_d(t)$ when $t \Rightarrow \infty$. In addition, the calculation of the control signal should only be carried out when the error signal exceeds a threshold $q_{nom}$.

## 6. Control Design

Since the motor torque $\tau_m(t)$ is a function of the currents flowing through the winding, the control loop will not only be closed by measuring the speed of the vehicle (or engine), but will also depend on the speed of the vehicle's current loop. In this way, the control algorithm will be implemented under an "internal loop-inner loop" configuration, as shown in Figure 7.

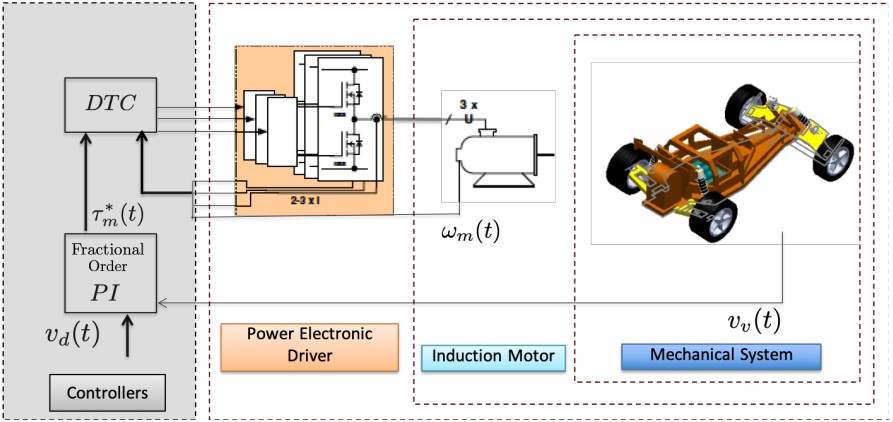

**Figure 7.** Block diagram of the control system.

The internal control loop is carried out by means of the well-known DTC [2,26]. As discussed before, DTC has the objective of ensuring that the IM delivers an adequate torque $\tau_m(t)$, so that together with the drive train, they generate the pulling force $F_t(t)$ that overcomes the forces due to gravity, the wind, the friction of bearing and the inertial effects and then maintain the desired speed $v_d$.

The external control loop is in charge of determining the desired torque profile $\tau *_m(t)$ that is used as a reference within the DTC, and which in turn will allow maintaining a constant desired speed vd.

## 7. Results of the Simulation

The DTC control can be implemented through simulation, including in the system the equations of both the dynamics of the EV and the IM. The electrical vehicle and the induction motor are described using Simulink blocks, which were built specifically for this job. First, step and ramp signals were tested as input signals (reference velocities), as seen in Figures 8 and 9. As was previously mentioned, both controllers were tuned to optimize the Integral of time-weighted absolute error (ITAE) signal of the car speed. As seen in Figure 8a,b the system under an FO-PI controller shows a bigger overshot when a step (0 to 25 km/h) is applied at 10 s of simulation. However, as is shown in Figure 8a, it has a lower ITAE value. Figure 8c,d show the torques of the system when this step is applied; both torque signals have similar maximum and minimum values.

Figure 9a,b show the output signals of the system, under PI and FO-PI controllers, when a ramp is applied at 140 s of simulation, the ramp changes the output from 0 to 15 km/h in 110 s. As seen in Figure 10 the response from the system under an FO-PI controller has a lower ITAE value. Figure 9c,d show the torques of the system when this ramp is applied; both torque signals are similar. The results shown in Figures 8–10 can be explained taking into account that the FO-PI controller has one more degree of freedom, because the integral action is of fractional order.

In the second test, a driving cycle is a speed profile drawn in a speed-time plane. This cycle is the "European driving cycle", which represents a typical way of driving, taking into account vehicle technology, traffic characteristics, roads, climatic characteristics and geography and also characteristics of the same drivers. These driving cycles are of great importance, among other purposes, in the development of technology for new cars and in the case of electric vehicles in the validation of models that predict the behavior of EVs and their energy consumption on the public road. In order to verify the correct performance of the simulation developed in this work, it was carried out in MATLAB/Simulink. Tables 3 and 4 show the parameters used in the simulation.

**Table 3.** Parameters of the induction motors (IM).

| Parameters of the Motor | Value |
| --- | --- |
| Poles | 4 |
| Rated power | 1.1 kW (Y: 380 V/4.43 A) |
| Nominal speed | 1415 rpm = 148.17 rad/s |
| Nominal Flux | 0.96 Wb |
| Nominal torque | 7.4 Nm |
| Rs | 9.21 Ω |
| Rr | 6.644 Ω |
| Lm | 0.44415 H |
| Ls | 0.03207 |
| Lr | 0.00847 H |

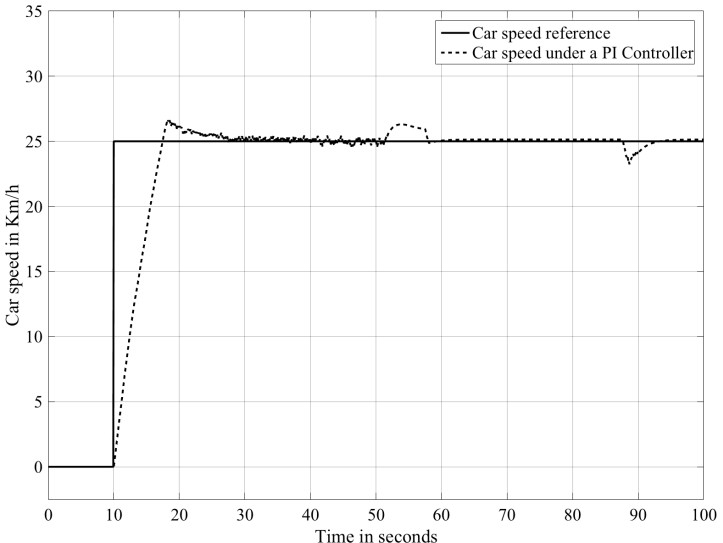

(**a**) Output under the proportional and integral (PI) controller.

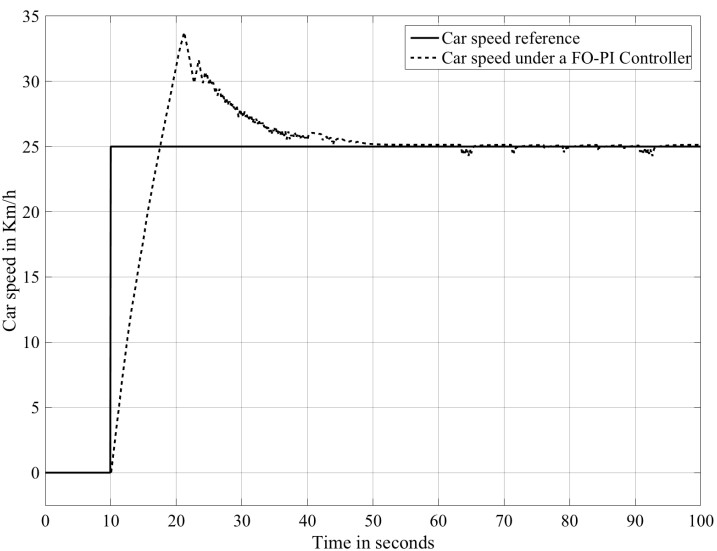

(**b**) Output under the fractional order (FO)-PI controller.

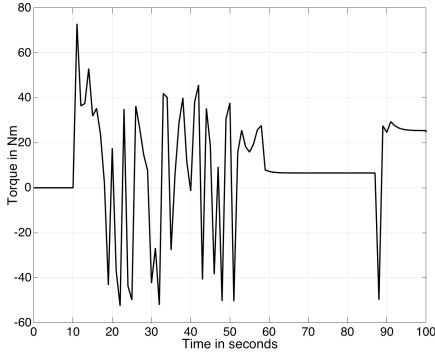

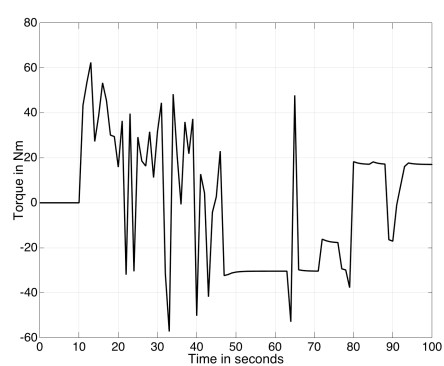

(**c**) Torque under the PI controller.

(**d**) Torque under the FO-PI controller.

**Figure 8.** Step response of the system.

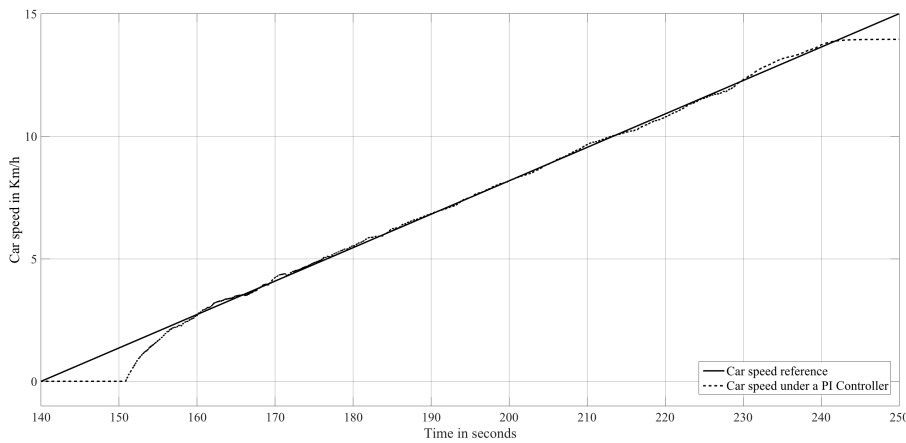

(**a**) Output under the PI controller.

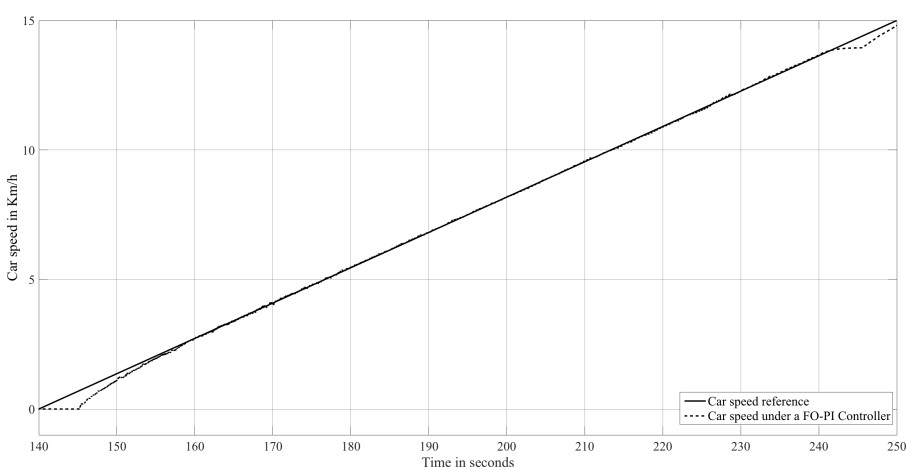

(**b**) Output under the FO-PI controller.

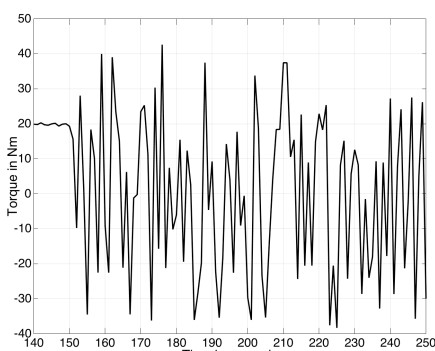

(**c**) Torque under the PI controller.

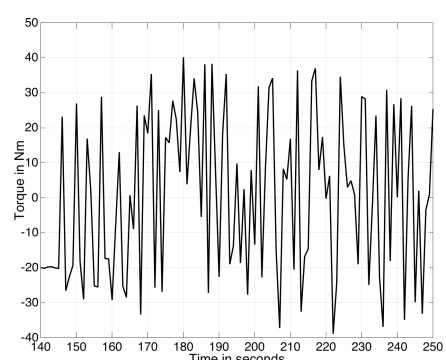

(**d**) Torque under the FO-PI controller.

**Figure 9.** Ramp response of the system.

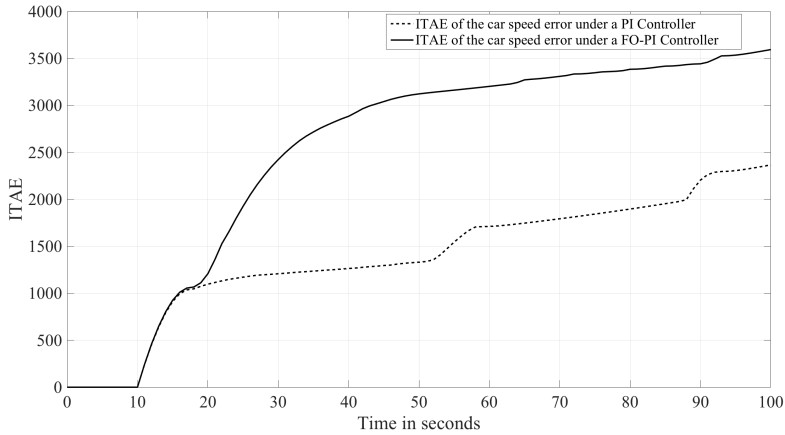

(**a**) Integral time-weighted absolute error (ITAE) step.

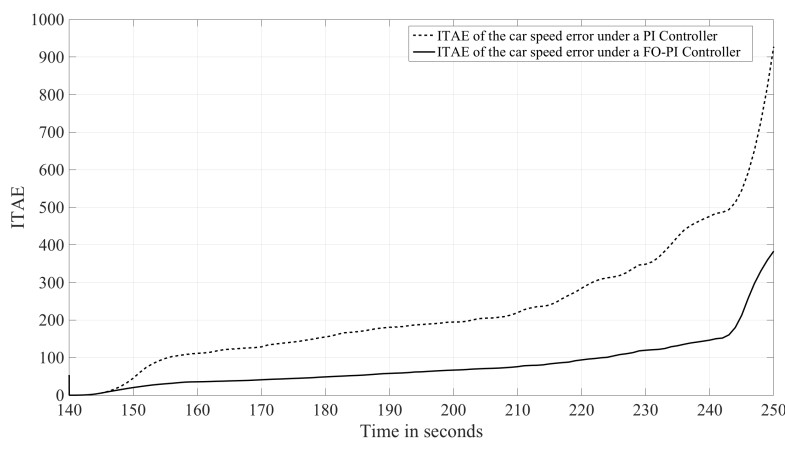

(**b**) ITAE ramp.

**Figure 10.** Comparison of the system under PI and FO-PI controller.

**Table 4.** Parameters of the electrical vehicle and the traction system.

| Parameters | Value |
|---|---|
| Vehicle mass | 200 Kg |
| Aerodynamic coefficient | 0.2 |
| Front Area | 1.5 m$^2$ |
| Coefficient of friction | 0.015 |
| Transmission efficiency | 0.97 |
| Tire radius | 0.2 m |
| Transmission ratio | 6 |

As was mention before, the desired speed profile was obtained from the European driving cycle, however it was scaled at a maximum speed of 30 km/h [27]. Figures 11 and 12 show the desired speed profile (Europe cycle) and the speed obtained by the vehicle, which follows the reference. The cycle has four repetitions and each of them has four stops, three accelerations, four periods of constant speed and four decelerations. The most representative values of the cycle are:

Average speed = 10 km/h
Total time = 14 min
Distance covered = 2 km 700 m
Maximum speed reached = 30 km/h

As shown in Figures 11 and 13, both controllers follow really close the speed reference, however if the ITAE of the speed error is analyzed, Figure 14, it is clear that the FO-PI controller has a lower value of this index. Figures 12 and 15 show the motor torque under both controllers respectively, as seen in the motor torque under the FO-PI controller which is lower that the one obtained by the classic PI.

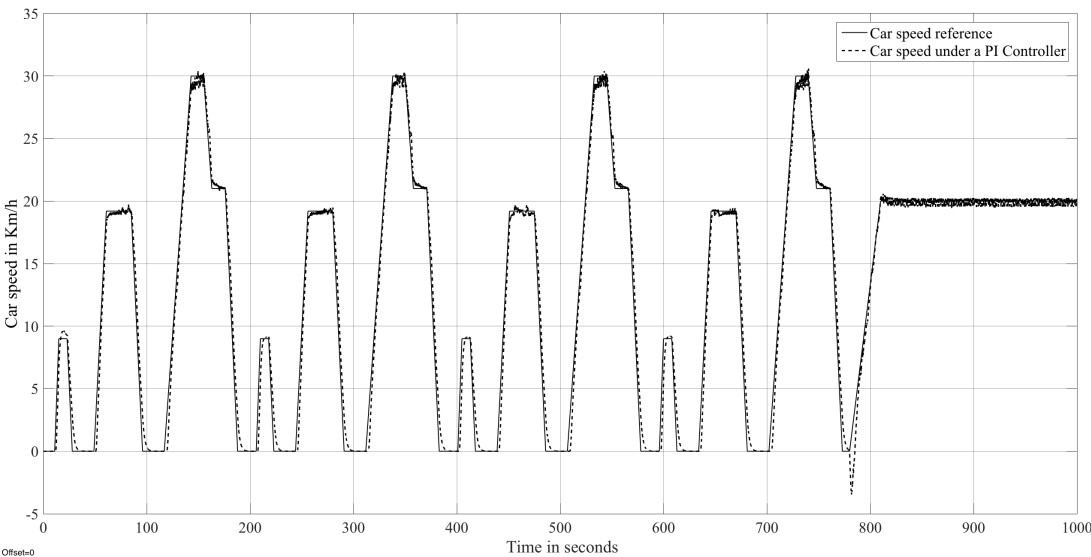

**Figure 11.** Car speed with the system under a PI controller.

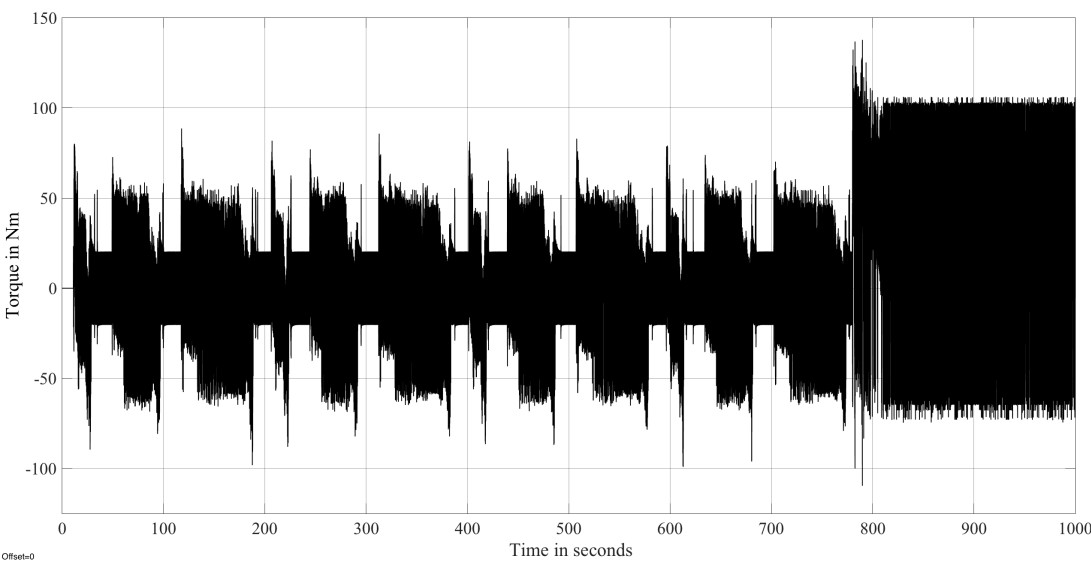

**Figure 12.** Motor torque with the system under a fractional order PI controller.

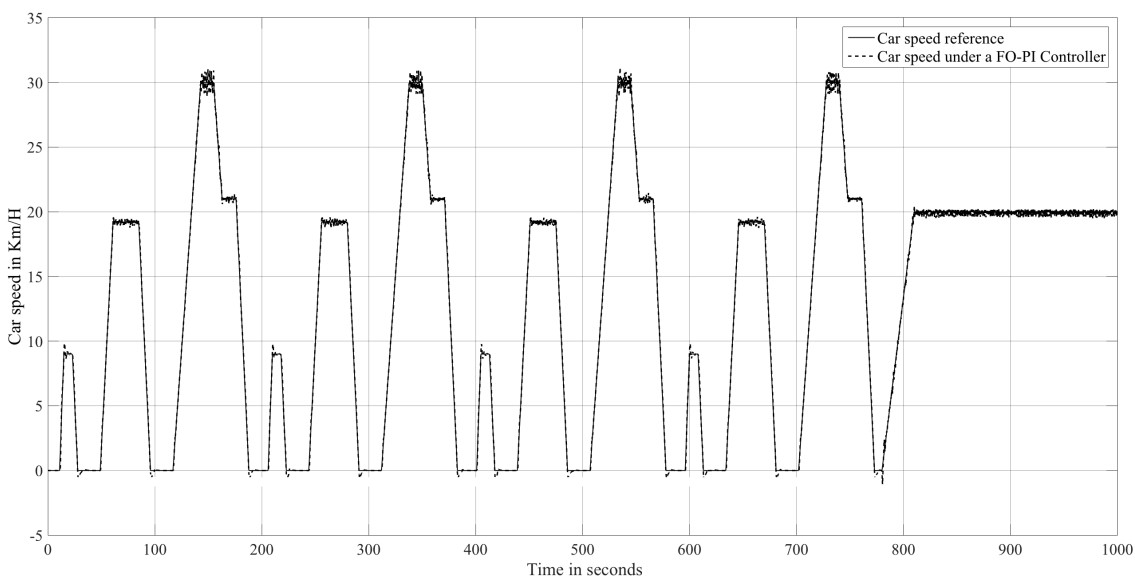

**Figure 13.** Car speed with the system under a fractional order PI controller.

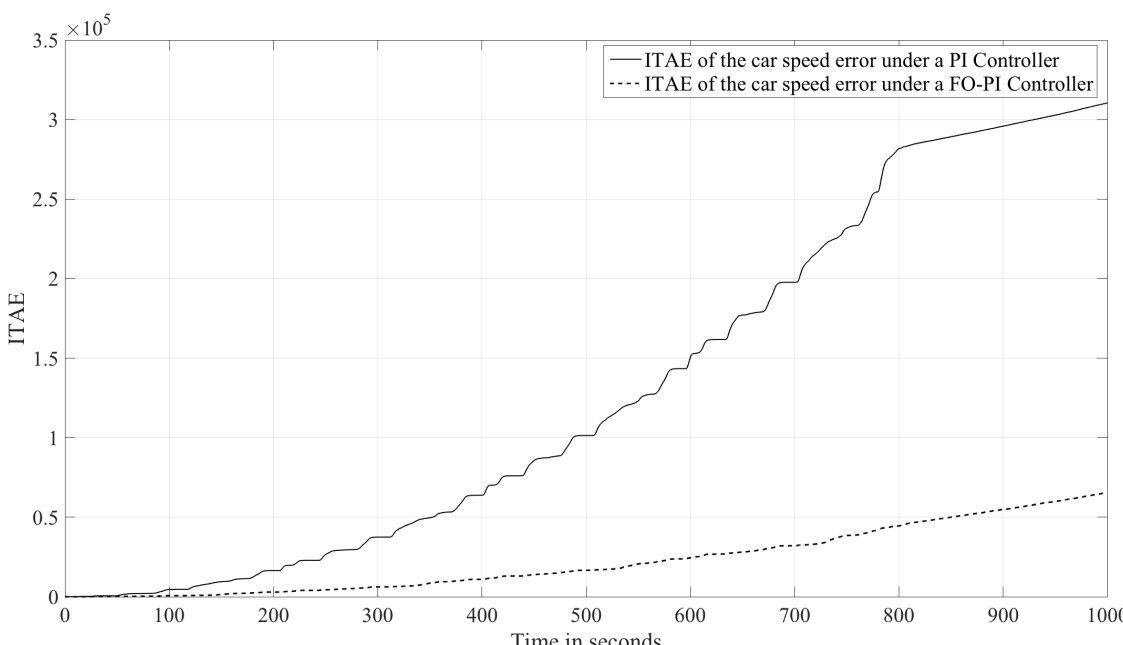

**Figure 14.** ITAE of the car speed with the system under both controllers.

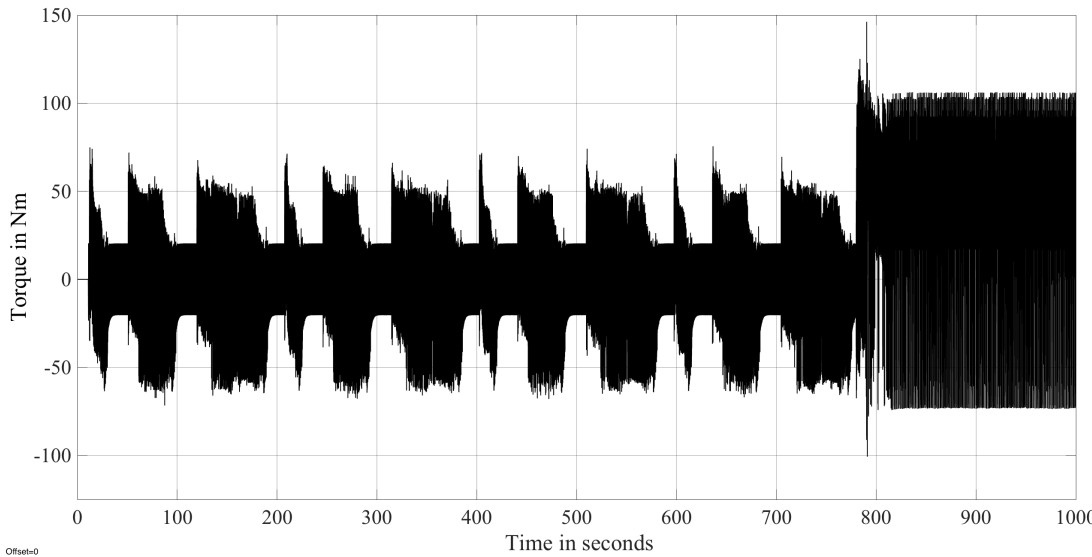

**Figure 15.** Motor torque with the system under a PI controller.

To increase the advantages to employ FO-PI controllers in this application, a third test was proposed, in this one a Least Squares based identification algorithm was used to get a model of the plant for a specific task and to tune the FO-PI controller. The driving cycle can be divided in position (step) and tracking (ramp) response. For those responses, Table 5 resumes the parameters of the second-order models obtained. Table 5 also shows the parameters of the FO-PI controller, which were calculated using the tuning method described in a previous section.

**Table 5.** Model and control parameters for step and ramp response.

| Response | Second-Order Model | | Control Parameters | | |
|---|---|---|---|---|---|
| | $\omega_n$ | $\zeta$ | $\lambda$ | $K_p$ | $K_i$ |
| Step | 2.1314 | 6.0171 | 0.15 | 3 | 1.5 |
| Ramp | 4.1964 | 3.3135 | 0.25 | 2 | 1.25 |

As can be seen in Figure 16, there is better performance when the FO-PI controller is tuned for the specific task that the system is executed. For this simulation, the speed reference has an initial value of zero then at 10 s of simulation time a step of 25 km/h is applied, followed by a decrease, at 100 s of simulation, of the reference until zero at 140 s of simulation. Finally a ramp is applied to elevate the reference from zero (150 s of simulation) to 15 km/h (250 s of simulation). As shown, the maximum peak is lower in the Figure 14a that is the case when the system is under the FO-PI controller tuned for step response. However the system follows the ramp in the better way in Figure 14b that is the case when the FO-PI controller was tuned for tracking response. Those results can be seen more clearly when the ITAE of the speed error is analyzed as shown in Figure 17. Figure 18 shows the motor torque under FO-PI controllers, tuned respectively for step and ramp response, as can be seen the motor torque under the FO-PI controller tuned for step response has a higher amplitude value that the one obtained with the same controller but tuned for ramp response, that is because the proportional and integral gains are higher than the case of step response.

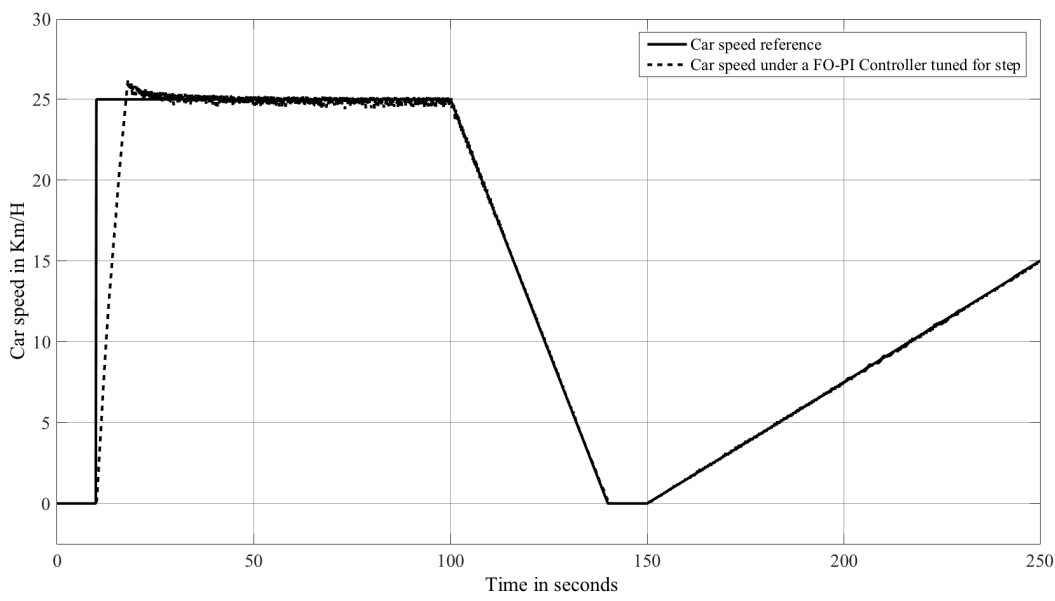

(**a**) Controller tuned for step response.

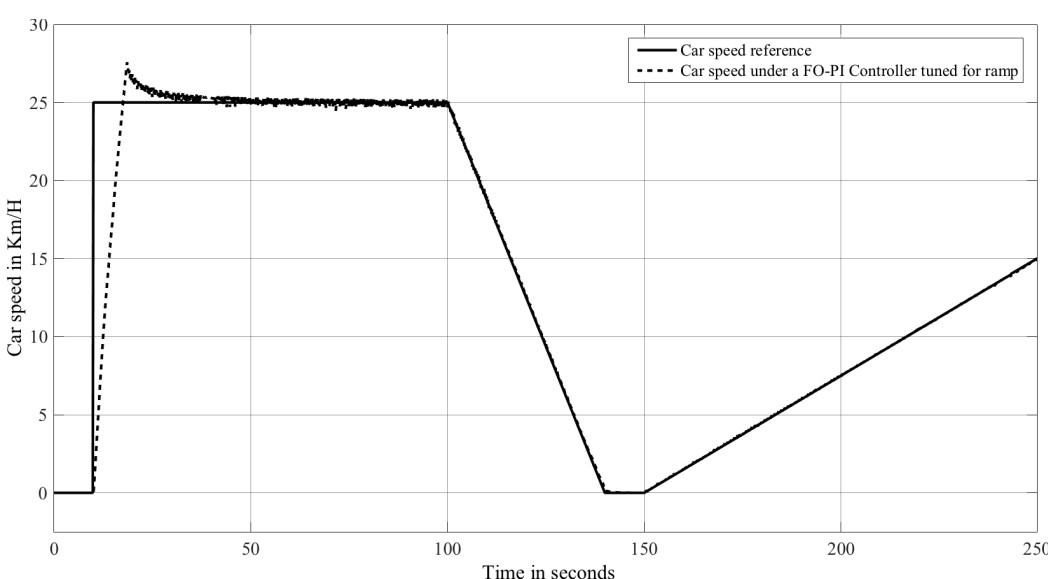

(**b**) Controller tuned for tracking response.

**Figure 16.** Car speed with the system under a Fractional Order PI controller.

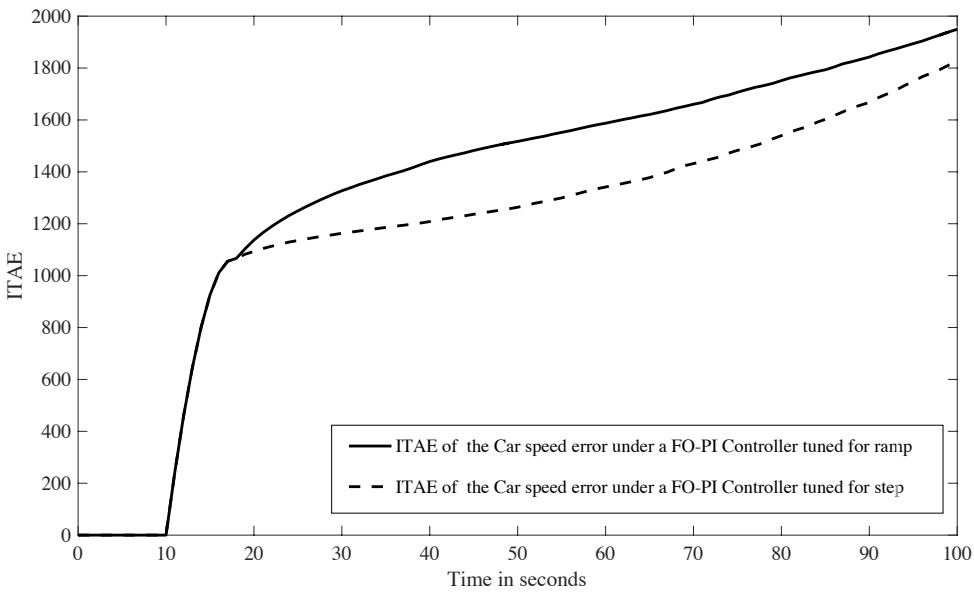

(**a**) Step response.

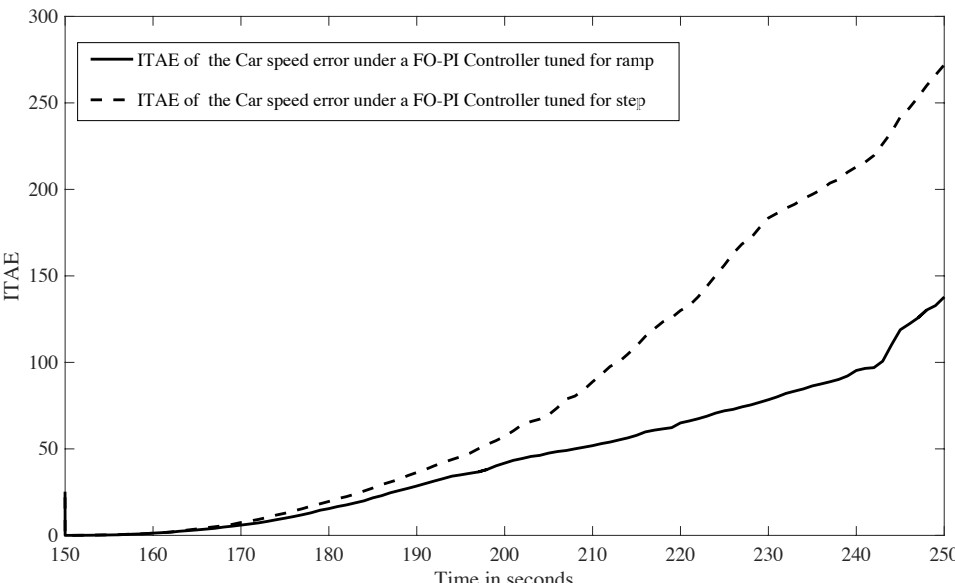

(**b**) Tracking response.

**Figure 17.** ITAE of the car speed with the system under FO-PI controller.

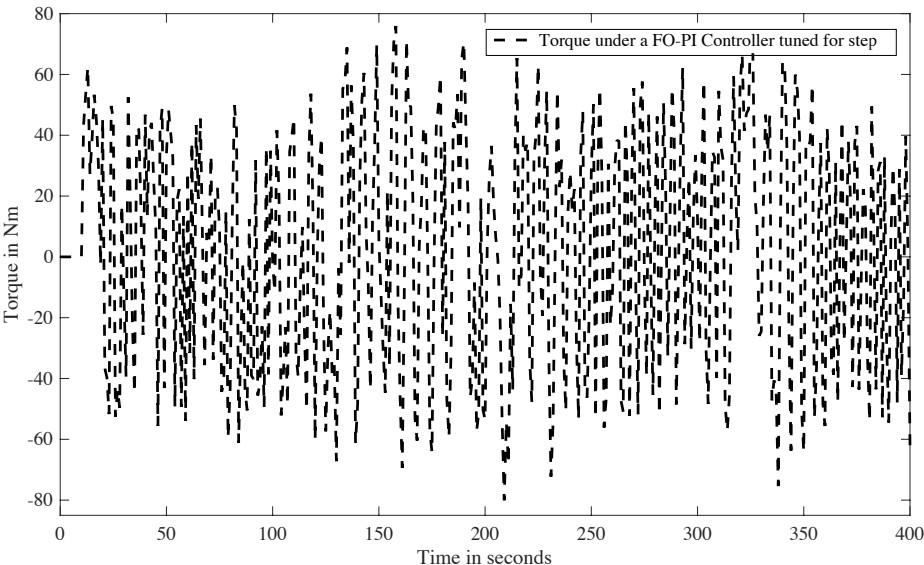

(**a**) Controller tuned for step response.

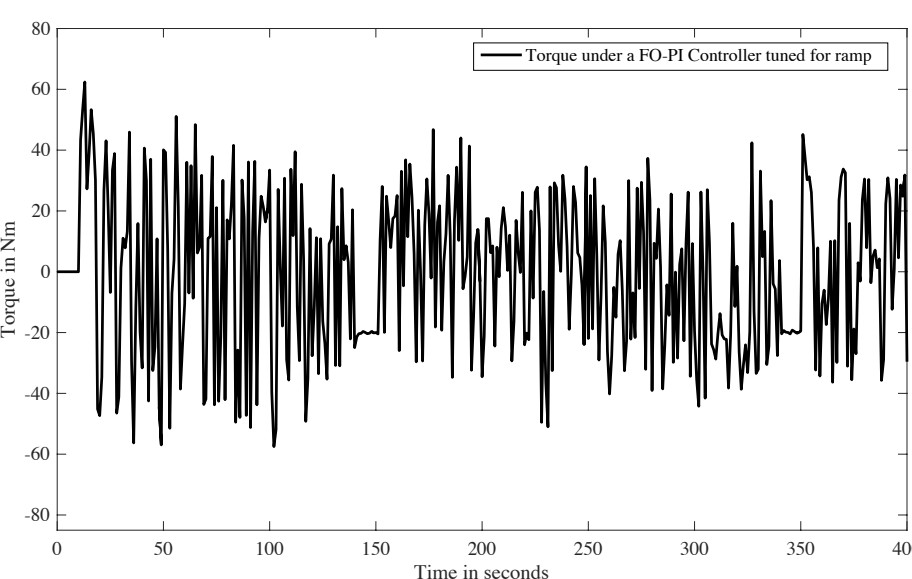

(**b**) Controller tuned for tracking response.

**Figure 18.** Motor torque with the system under a FO-PI controllers.

## 8. Discussion

In this paper, a DTC of the traction system of an EV was addressed. The traction system is driven by an IM using a PI controller of fractional order. Torque and motor flux controllers are the basis for the DTC. Besides, the stator flux angle is also calculated. Look-up and direct look-up tables were employed to calculate the flux vector and the optimum switching vector applied to the motor.

The New Europe Drive Cycle (NEDC) was used in simulations to determine speed profiles; these are the references to assess the performance of the presented technique. Derived from these results, it can be stated that the control satisfies the demands satisfactorily, which results in a friendly speed control. Stator flow and motor torque are also kept under control through space voltage vectors. Likewise, the evaluated results have shown that the instantaneous flux vector of the stator and the

electromagnetic torque originated in an IM can be estimated directly using the currents and voltages measured at the motor terminals.

The modeling process, control design and simulation of the electric vehicle (EV) traction system were discussed. The results show that an FO-PI Controller increases the performance of the system. In the future, a real-time implementation is expected.

**Author Contributions:** G.A.M.-H, G.M.-A. and J.F.G.-C. developed the theoretical framework and G.A.M.-H. and E.P.-S. performed the simulation results. All authors have read and agreed to the published version of the manuscript.

**Acknowledgments:** German Ardul Munoz-Hernandez wishes to thank the Instituto Tecnologico de Puebla, who has supported him with a sabbatical year that made the participation in this work possible.

**Conflicts of Interest:** The authors declare no conflict of interest.

## Abbreviations

The following abbreviations are used in this manuscript:

| | |
|---|---|
| EV | Electrical vehicle |
| HEV | Hybrid electrical vehicle |
| ICE | Internal combustion engines |
| IM | Induction motor |
| DTC | Direct torque control |
| PID | Proportional, integral and derivative |
| FO-PI | Fractional order proportional and integral |
| NEDC | New Europe Drive Cycle |
| ITAE | Time-weighted absolute error |
| BUAP | Meritorious Autonomous University of Puebla |

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
