# Peer review of "Fractional Order PI-Based Control Applied to the Traction System of an Electric Vehicle (EV)"

_applsci, doi:10.3390/app10010364_

Round 1
Reviewer 1 Report
You should detail the tuning of the fractional PI controller.
Author Response
Thank you for your comments, which have allowed us to improve the manuscript. In the following lines, your observations are answered:
Remark. You should detail the tuning of the fractional PI controller.
Answer. A deep discussion about how the fractional controller is approximated in this work has been added to the paper, pages 10 and 11.
Reviewer 2 Report
This article is focusing on designing cruise control based on a fractional-order PI Direct Torque Control applied to the traction system of an electric vehicle(Cruise Control). There are some major concern regarding the paper as follows:
1-Authors did not explain solution for problems which could be presented such as starting requirement of flux and torque estimator.
2-There are many errors and weak structures in writing this paper. Authors should read the whole article again and reconsider the English language fluency of their paper.
3- Figure 8 is very small and unreadable.
Author Response
Thank you for your comments, which have allowed us to improve the manuscript. In the following lines, your observations are answered:
Remark. Authors did not explain solution for problems, which could be presented such as starting requirement of flux and torque estimator.
Answer. A deep discussion about the starting requirement of flux and torque estimator has been included in paper, pages 6 and 7.
Remark. There are many errors and weak structures in writing this paper. Authors should read the whole article again and reconsider the English language fluency of their paper.
Answer. The manuscript has been completely reviewed.
Remark. Figure 8 is very small and unreadable.
Answer. The fonts on Figures 8 to 11 and 13 have been resizes.
Reviewer 3 Report
The paper proposes the a DTC of the traction system of an EV by an IM using a PI controller of fractional order. The modelling, simulation, control and discussion have been presented by the authors in a standard form. In my view, the paper is not suitable for publication in applied sciences respect to the following drawbacks:
1- My main concern is about the contribution of the manuscript. The authors have used an off-the-shelf method for a straightforward engineering problem. The controller has been applied to the developed model and the results are presented. I cannot see any contribution in this paper to consider it as a research article.
2- The objective of the literature review should be a gap analysis to justify the necessity of doing this piece of research. What shortcomings/rooms have been addressed by this research and how? What is the relative merits of the proposed method to the other publicly available methods and approaches?
3- The modelling and the system is very simple and straightforward. Is there any contribution in this section?
4- Why and how this type of control approach has been selected for the in-hand problem? What performance indices have been taken into account?
5- The result analysis section must be re-written in a professional feature. Much more details, discussions, and comparisons are required to illustrate the effectiveness of the proposed approach.
6- What are the main limitations, assumptions, and constraints of the used method.
7- Figures 8-10 should be replaced with higher quality ones and more details and discussions should be added about them.
Author Response
Thank you for your comments, which have allowed us to improve the manuscript. In the following lines, your observations are answered:
Remark. The objective of the literature review should be a gap analysis to justify the necessity of doing this piece of research. What shortcomings/rooms have been addressed by this research and how? What is the relative merits of the proposed method to the other publicly available methods and approaches?
Answer. Although there are publications where similar results have been reported that are similar to certain parts of this work, this paper integrates the models of the induction motor and the car, for a particular application, where two types of control are tested.
Remark. The modelling and the system is very simple and straightforward. Is there any contribution in this section?
Answer. The model used for the electric car is useful for the scope of this work. As the reviewer comments, this is simple however for this stage of the work is sufficient. Nonetheless, for future work this should consider other aspects related to the dynamics of the car.
Remark. Why and how this type of control approach has been selected for the in-hand problem? What performance indices have been taken into account?
Answer. The control that was proposed for this problem initially was a classic PI control. However, since the integral index that was pursued to reduce was not within the limits wanted, the possibility of using a different control was analyzed. Since the fractional operators of the FO-PID increases by two degrees of freedom the controller (in the specific case of FO-PI one degree), this controller was considered in this work. On the other hand, the way in which the control is being carried out does not substantially increase the calculation time, but substantially improving the integral absolute error index (ITAE), as is explain in the Paper (section 4 “Fractional Order Control”, pages 10 and 11.
Remark. The result analysis section must be re-written in a professional feature. Much more details, discussions, and comparisons are required to illustrate the effectiveness of the proposed approach.
Answer. The manuscript has been completely reviewed.
Remark. What are the main limitations, assumptions, and constraints of the used method?
Answer. Like all controllers, FO-PI has a high dependence on the parameters selected in its tuning. In this work the ITAE index was used, that means that if the system is analyzed for another type of performance, it could be that the system achieved does not behave "optimally" under that “new” index. Another problem that arises, related to the approximation of the fractional control, is the dependence of the controller on the working frequency. That is due to the approximation of the fractional operators is made by a led/lead compensator, that is discussed in the section 4 of the manuscript, “Fractional Order Control”, pages 10 and 11.
Remark. Figures 8-10 should be replaced with higher quality ones and more details and discussions should be added about them.
Answer. The fonts on Figures 8 to 11 and 13 have been resizes.
Reviewer 4 Report
The authors have done quite well to address all the concerns and comments of reviewers.
Round 2
Reviewer 2 Report
The article now looks better and could be considered for publication.
Reviewer 3 Report
The authors have addressed some of my concern into some extend. However, I still believe that there is no originality/contribution in the paper to be considered as a research article in a high profile journal like Applied Sciences. So, I am forced to persist on my previous decision on the rejection of the paper.